# *Mycobacterium tuberculosis* Infection among 1,659 Silicosis Patients in Zhejiang Province, China

Qingluan Yang,[a] Miaoyao Lin,[b] Zhangyufan He,[a] Xuefeng Liu,[b] Yuzhen Xu,[a] Jing Wu,[a] Feng Sun,[a] Tian Jiang,[b] Yan Gao,[a] Xitian Huang,[b] Wenhong Zhang,[a,c,d] (ID) Qiaoling Ruan,[a] Lingyun Shao[a]

[a]Department of Infectious Diseases, Shanghai Key Laboratory of Infectious Diseases and Biosafety Emergency Response, National Medical Center for Infectious Diseases, Huashan Hospital, Fudan University, Shanghai, China
[b]The First People's Hospital of Wenling, Zhejiang, China
[c]National Clinical Research Center for Aging and Medicine, Huashan Hospital, Fudan University, Shanghai, China
[d]Key Laboratory of Medical Molecular Virology (MOE/MOH), Shanghai Medical College, Fudan University, Shanghai, China

Qingluan Yang, Miaoyao Lin, and Zhangyufan He contributed equally to this article. The order was determined by the corresponding author after negotiation.

**ABSTRACT** Silicosis is a well-established risk factor for *Mycobacterium tuberculosis* infection. This study aimed to estimate the burden and risk factors of *M. tuberculosis* infection. Silicosis patients from Zhejiang Province were screened for *M. tuberculosis* by sputum culture, chest radiographs, whole-blood gamma interferon (IFN-$\gamma$) release assay (QuantiFERON-TB Gold In-Tube [QFT-GIT]), and tuberculin skin test (TST). Potential risk factors for *M. tuberculosis* were identified. Data for 1,659 patients were obtained from 1,684 participants. Of these, 1,656 (99.8%) were men, and the average age was 58 (54 to 63) years. The prevalence of active tuberculosis (ATB) was 6,340/100,000 (6.34%) people; the proportion of patients with latent tuberculosis infection (LTBI) was 50.6%. Age (odds ratio [OR] = 1.059, 95% confidence interval [CI] = 1.020 to 1.099, $P$ = 0.003), being underweight (OR = 2.320, 95% CI = 1.057 to 5.089, $P$ = 0.036), and having a history of exposure to TB patients (OR = 4.329, 95% CI = 1.992 to 9.434, $P$ < 0.001) were associated with ATB; BCG vaccination could reduce ATB risk in silicosis patients (OR = 0.541, 95% CI = 0.307 to 0.954, $P$ = 0.034). Among patients without ATB, the QFT-GIT positivity rate was 40.5%, which was affected by silicosis severity, while that of TST was 57.2%. BCG vaccination was an independent factor for LTBI risk reduction (OR = 0.612, 95% CI = 0.468 to 0.801, $P$ < 0.001). The quantitative results of QFT-GIT decreased with silicosis stage ($H$ = 6.037; $P$ = 0.048). In conclusion, *M. tuberculosis* prevalence was high in silicosis patients. BCG vaccination reduced the risk of both ATB and LTBI in silicosis patients.

**IMPORTANCE** This study evaluated the prevalence of *Mycobacterium tuberculosis* infection in silicosis patients in mainland China and identified the potential risk factors for both active tuberculosis (ATB) and latent tuberculosis infection (LTBI). We believe that our study makes a significant contribution to the literature because we demonstrated that *M. tuberculosis* prevalence was high among silicosis patients. BCG vaccination was an independent factor that reduced the risk of *M. tuberculosis* infection in patients with silicosis. Furthermore, we show that the prevalence of LTBI in patients with silicosis may have been underestimated by immunological detection methods. This study can help to identify targeted subgroups prioritized for *M. tuberculosis* control and to reduce the risk of disease development.

**KEYWORDS** silicosis, *Mycobacterium tuberculosis*, tuberculosis, latent tuberculosis infection, prevalence, QuantiFERON-TB Gold In-Tube, BCG vaccination

Address correspondence to Lingyun Shao, lingyun26@fudan.edu.cn, or Qiaoling Ruan, ruan_qiao_ling@163.com.

The authors declare no conflict of interest.

Silicosis is a fibrotic lung disease caused by inhalation of crystalline silicon dioxide or silica and is one of the most important occupational diseases worldwide (1). Twenty years ago, China had the highest incidence of silicosis, with 6,000 new cases

and >24,000 deaths reported annually (2). There are still more than 120,000 new cases and >7,000 deaths annually (3). More than 23 million workers are exposed to crystalline silica in China (4). The relationship between silicosis and tuberculosis (TB) has long been recognized. Exposure to silica dust or the development of silicosis predisposes individuals to TB (1). In developing countries, mineral mining (particularly gold mining) may contribute to national TB rates (5). The risk of TB remains high for gold miners in South Africa, even after they are no longer exposed. TB risk increases with silicosis severity and is high in patients with acute and accelerated silicosis (6, 7). Latent tuberculosis infection (LTBI) is another problem in silicosis patients (8). The diagnosis of LTBI depends mainly on the host immune reaction rather than the bacterium itself. The tuberculin skin test (TST) and gamma interferon (IFN-γ) release assay are two methods frequently used for diagnosis. In LTBI, there are no clinical manifestations, but 5 to 15% of these patients will experience reactivation of TB during their lifetime. Silicosis contributes to this high TB reactivation rate. Among silicosis patients with a TST result of ≥10 mm, the TB rate was 36.9/1,000 person-years (95% confidence interval [CI] = 28.2 to 48.1), and the World Health Organization (WHO) recommends both testing and preventative treatment for LTBI in silicosis patients.

In the present study, we investigated the prevalence of both active tuberculosis (ATB) and LTBI in silicosis patients in Zhejiang Province, China. We sought to characterize these populations and identify the risk factors associated with *M. tuberculosis* infection, including ATB and LTBI.

## RESULTS

**Participant characteristics.** A total of 1,684 silicosis patients who were recruited from Wenling and Sanmen Counties participated in the investigation in 2014 and 2018. Sixteen participants without chest radiograph confirmation and nine with missing medical histories were excluded from the analysis. Of the 1,659 silicosis patients, 98 were diagnosed with ATB, and 349 had a history of TB. Among the patients excluded from ATB, 591 patients did not undergo immunological examinations for *M. tuberculosis* infection, including those who had non-tuberculosis mycobacterial infections. A total of 970 cases underwent QFT-GIT examination, 393 of which were positive, and the positive rate of QFT-GIT was 40.5%. In addition, 409 cases of exposure to silica dust underwent TST examination; 235 of these cases were positive, and the TST positive rate was 57.2%. A total of 491 patients were diagnosed with LTBI, with positive results for either QFT-GIT or TST. Therefore, 98 ATB, 491 LTBI, and 479 non-TB patients were included in the analysis (Fig. 1). The *M. tuberculosis* infection status for patients in different silicosis categories is given in Fig. S1.

The main baseline characteristics of the study population are presented in Table 1. Most were men, and the median (interquartile range [IQR]) age was 58 (IQR = 54 to 63) years. More than half were normal weight, and 7.5% were underweight. The distribution of silicosis categories showed that nearly half of the participants (826 [49.8%]) had stage 3 disease. Regarding education, 28.1% (466/1,659) were illiterate, and 56.3% (934/1,659) had primary school education. The most common occupation by industry type was quarries, followed by tunnels. Three-quarters (1,246/1,659) of participants had ever smoked, and 21.7% (360/1,659) had a history of previous TB. Nearly 30% (490/1,659) of participants showed a BCG (bacillus Calmette-Guérin) scar. Symptoms associated with silicosis and tuberculosis, such as cough, dyspnea, chest pain, and wheezing, were present in most patients with silicosis. In addition, some patients had fever, weight loss, and night sweats.

**Risk factors for active tuberculosis.** Among the 1,545 patients who successfully completed sputum culture, 98 cases of ATB were found, indicating that the prevalence of TB in this population was 6,340/100,000 (6.34%). The association analyses that identified the potential factors related to ATB are listed in Table 2. In both univariate and multivariate analyses, ATB was associated with age ($P = 0.003$, adjusted odds ratio [OR] = 1.059 [95% CI = 1.020 to 1.099]). Being underweight and having contact with TB patients were associated with an increased risk of ATB, with adjusted OR values of

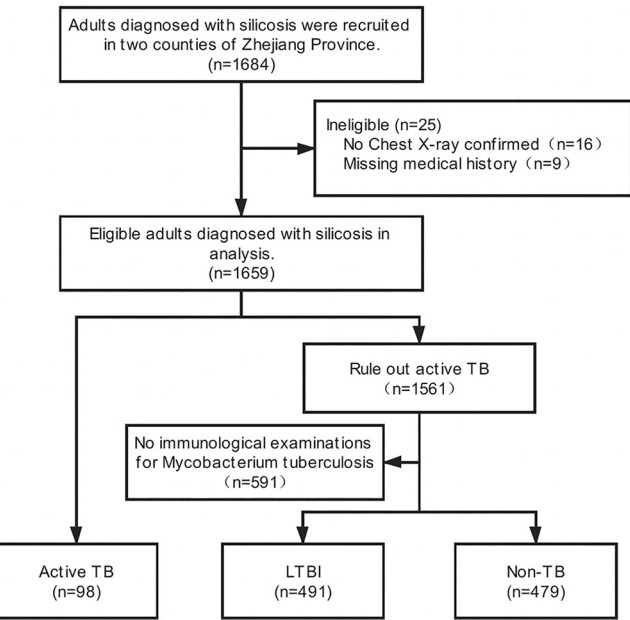

**FIG 1** Flow chart for selection of study population.

2.320 (95% CI = 1.057 to 5.089) and 4.329 (95% CI = 1.992 to 9.434), respectively. In contrast, having a BCG scar reduced the risk of ATB, with an adjusted OR of 0.541 (95% CI = 0.307 to 0.954). Patients with symptoms of wheezing, fever, and weight loss were more likely to have ATB, but there was no significant association in the multivariate analysis. Interestingly, no significant association was found between silicosis category, education, cigarette smoking status, drinking status, and ATB in our study population.

**Risk factors for latent tuberculosis infection.** A total of 970 silicosis patients who did not have ATB were tested using QFT-GIT or TST, and 491 were found to have LTBI, with a percentage as high as 50.6%. To identify the potential factors associated with LTBI in silicosis patients, univariate and multivariate analyses were also conducted between the LTBI and non-TB groups, as shown in Table 3. Silicosis stage 2 (compared to stage 0) was associated with LTBI, with an adjusted OR of 0.527 (95% CI = 0.320 to 0.868). Having a BCG scar reduced the risk of LTBI, with an adjusted OR of 0.791 (95% CI = 0.692 to 0.904). Patients who were younger or had a higher BMI appeared to be more vulnerable to LTBI, but no significant differences were found. Neither cigarette smoking nor drinking status was associated with LTBI.

**QFT-GIT quantitative result distribution among different silicosis categories and age groups.** As shown in Fig. 2A, among the QFT positives, the median QFT quantitative results decreased significantly with increasing silicosis category ($H = 6.037$, $P = 0.048$). To verify the immune response capacity of lymphocytes in patients with various categories of silicosis, the proportions of leukocytes and lymphocytes in the peripheral blood of patients were compared. The results showed that the higher the stage of silicosis, the higher the number of leukocytes and the lower the proportion of lymphocytes ($H = 47.41$, $P < 0.0001$). In addition, the median QFT quantitative results among patients aged <65 and ≥65 years were different ($U = 4655$, $P = 0.049$) (Fig. 2B).

**Comparison of QFT-GIT and TST among silicosis patients.** Among the 409 silicosis patients for whom TST and QFT-GIT results were obtained, the agreement between TST (10-mm cutoff) and QFT-GIT was low (kappa coefficient, 0.35). When the TST results were divided into four groups according to the induration diameter, the corresponding median IFN-$\gamma$ level increased as QFT-GIT responses increased with TST responses ($H = 57.09$, $P < 0.001$) (Fig. 3A). The prevalence of LTBI by QFT-GIT and TST stratified by silicosis category is shown in Fig. 3B.

**TABLE 1** Baseline characteristics of the study populations[a]

| Variable | Total | ATB | LTBI | Non-TB |
|---|---|---|---|---|
| No. | 1,659 | 98 | 491 | 479 |
| Median age in yrs (IQR) | 58 (54–63) | 59 (55–64) | 57 (53–61) | 57 (53–61) |
| No. male (%) | 1,656 (99.8) | 98 (100) | 491 (100) | 98 (100) |
| BMI, kg/m² (%) | | | | |
| <18.5 | 124 (7.5) | 14 (14.3) | 21 (4.3) | 24 (5.0) |
| ≥18.5 to <24.0 | 875 (52.7) | 56 (57.1) | 246 (50.1) | 247 (51.6) |
| ≥24.0 to <28.0 | 556 (33.5) | 23 (23.5) | 190 (38.7) | 172 (35.9) |
| ≥28 | 104 (6.3) | 5 (5.1) | 34 (6.9) | 36 (7.5) |
| Silicosis categories[b] (%) | | | | |
| 0 | 174 (10.5) | 6 (6.1) | 62 (12.6) | 37 (7.7) |
| 1 | 334 (20.1) | 13 (13.3) | 107 (21.8) | 90 (18.8) |
| 2 | 325 (19.6) | 17 (17.3) | 100 (20.4) | 127 (26.5) |
| 3 | 826 (49.8) | 62 (63.3) | 222 (45.2) | 225 (47.0) |
| Education (%) | | | | |
| Illiteracy | 466 (28.1) | 23 (23.5) | 138 (28.1) | 130 (27.1) |
| Primary school | 934 (56.3) | 64 (65.3) | 266 (54.2) | 265 (44.5) |
| Middle school or higher | 259 (15.6) | 11 (11.2) | 87 (17.7) | 84 (17.5) |
| Main job (%) | | | | |
| Quarry | 1,451 (87.5) | 83 (84.7) | 433 (88.2) | 440 (91.9) |
| Tunnel | 133 (8.0) | 6 (6.1) | 32 (6.5) | 23 (4.8) |
| Others[c] | 75 (4.5) | 9 (9.2) | 26 (5.3) | 16 (3.3) |
| Quit working (%) | 1591 (95.9) | 95 (96.9) | 460 (94.5) | 454 (94.8) |
| Cigarette smoking (%) | | | | |
| Smoker | 409 (24.7) | 30 (30.6) | 166 (33.8) | 143 (29.9) |
| Ex-smoker | 837 (50.5) | 39 (39.8) | 216 (44.0) | 218 (45.5) |
| Never smoke | 413 (24.9) | 29 (29.6) | 109 (22.2) | 118 (24.6) |
| Drinking (%) | | | | |
| Drinker | 742 (44.7) | 49 (50.0) | 222 (45.2) | 205 (42.8) |
| Abstainer | 282 (17.0) | 16 (16.3) | 91 (18.5) | 77 (16.1) |
| Never drink | 635 (38.3) | 33 (33.7) | 178 (36.3) | 197 (41.1) |
| History of previous TB | 360 (21.7) | 9 (9.2) | 62 (12.6) | 29 (6.2) |
| Contact of TB patients | 93 (5.6) | 14 (14.3) | 21 (2.2) | 17 (3.5) |
| BCG scar | 490 (29.5) | 20 (20.4) | 157 (32.0) | 213 (36.9) |
| Manifestations (%) | | | | |
| Cough | 1,053 (63.5) | 71 (72.4) | 303 (62.2) | 303 (63.3) |
| Dyspnea | 769 (46.4) | 45 (45.9) | 224 (46.0) | 190 (39.7) |
| Chest pain | 321 (19.3) | 14 (14.3) | 85 (17.5) | 77 (16.1) |
| Wheeze | 916 (55.2) | 62 (63.3) | 241 (49.5) | 229 (47.8) |
| Fever | 67 (4.0) | 9 (9.2) | 18 (3.7) | 13 (2.7) |
| Loss of wt | 148 (8.9) | 15 (15.3) | 34 (7.1) | 34 (7.1) |
| Night sweat | 171 (10.3) | 13 (13.3) | 54 (11.1) | 40 (8.4) |
| Complications (%) | | | | |
| Diseases of the cardiovascular system[d] | 452 (27.2) | 27 (18.8) | 121 (24.6) | 117 (24.4) |
| Diseases of the respiratory system[e] | 144 (8.7) | 9 (9.2) | 40 (8.1) | 38 (7.9) |
| Diseases of the digestive system[f] | 293 (17.7) | 11 (11.2) | 77 (15.7) | 93 (19.4) |
| Laboratory examination, median (IQR) | | | | |
| WBC (×10⁹/L) | 6.8 (5.9–8.0) | 7.0 (5.5–8.0) | 6.8 (5.9–8.0) | 7.0 (6.0–8.0) |
| Ly (×10⁹/L) | 1.8 (1.4–2.3) | 1.5 (1.2–2.0) | 1.9 (1.5–2.4) | 1.9 (1.4–2.3) |
| Ne (%) | 64.3 (58.0–70.6) | 68.4 (61.9–84.1) | 62.8 (57.1–68.2) | 63.6 (57.3–71.4) |
| PLT (×10¹²/L) | 222 (189–256) | 224 (198–265) | 227 (195–260) | 218 (185–248) |
| Hb (g/L) | 145 (137–152) | 141 (132–150) | 146 (139–153) | 146 (139–153) |

[a]Data are presented as numbers of patients (%) or median values (interquartile range [IQR]) unless noted otherwise in column 1. BMI, body mass index; BCG, bacillus Calmette-Guérin; WBC, white blood cell; Ly, lymphocyte; Ne, neutrophil; PLT, platelet; Hb, hemoglobin.
[b]Silicosis categories were determined according to the 2011 revised edition of the ILO *Guidelines for the Use of the ILO International Classification of Radiographs of Pneumoconioses*.
[c]That is, including surface drillers/stone-splitters/drillers in the construction trade or quarries, plasterers, etc.
[d]Coronary disease or hypertension, etc.
[e]Chronic obstructive pneumonia or bronchial asthma, etc.
[f]Peptic ulcer or chronic hepatopathy, etc.

## DISCUSSION

This is the first large-scale epidemiological study conducted in patients with silicosis to estimate the *M. tuberculosis* burden and to investigate potential risk factors for *M. tuberculosis* infection in mainland China. The prevalence of ATB is as high as

**TABLE 2** Potential factors associated with active tuberculosis in the study population[a]

| Variable | P | OR (95% CI) | Adj P | Adj OR (95% CI) |
|---|---|---|---|---|
| Age | **0.007** | | **0.003** | 1.059 (1.020–1.099) |
| BMI (kg/m$^2$) | | | | |
| <18.5 | **0.017** | 2.573 (1.215–5.239) | **0.036** | 2.320 (1.057–5.089) |
| ≥18.5 to <24 | | REF | | REF |
| ≥24 to <28 | 0.059 | 0.590 (0.355–1.003) | 0.054 | 0.583 (0.336–1.010) |
| ≥28 | 0.390 | 0.613 (0.251–1.579) | 0.513 | 0.714 (0.260–1.961) |
| Silicosis categories[b] | | | | |
| 0 | | REF | | |
| 1 | 0.794 | 0.89 (0.318–2.535) | | |
| 2 | 0.792 | 0.83 (0.297–2.218) | | |
| 3 | 0.314 | 1.699 (0.725–3.978) | | |
| Education | | | | |
| Illiteracy | | REF | | |
| Primary school | 0.255 | 1.365 (0.809–2.262) | | |
| Middle school or higher | 0.569 | 0.74 (0.329–1.624) | | |
| Main job | | | | |
| Quarry | **0.023** | 0.335 (0.151–0.824) | 0.217 | 0.567 (0.230–1.396) |
| Tunnel | 0.239 | 0.464 (0.144–1.453) | 0.258 | 0.478 (0.133–1.718) |
| Others[c] | | REF | | REF |
| Quit working | 0.365 | 1.744 (0.516–5.893) | | |
| Cigarette smoking status | | | | |
| Smoker | 0.665 | 0.854 (0.494–1.480) | | |
| Ex-smoker | 0.269 | 0.728 (0.424–1.242) | | |
| Never smoke | | REF | | |
| Drinking | | | | |
| Drinker | 0.182 | 1.427 (0.888–2.296) | | |
| Abstainer | 0.499 | 1.24 (0.627–2.322) | | |
| Never drink | | REF | | |
| Contact of TB patients | **<0.001** | 4.529 (2.150–9.524) | **<0.001** | 4.329 (1.992–9.434) |
| BCG scar | **<0.001** | 0.320 (0.190–0.540) | **0.034** | 0.541 (0.307–0.954) |
| Manifestations | | | | |
| Cough | 0.083 | 1.527 (0.945–2.470) | | |
| Dyspnea | 0.251 | 1.291 (0.834–2) | | |
| Chest pain | 0.658 | 0.87 (0.47–1.611) | | |
| Wheeze | **0.005** | 1.88 (1.201–2.943) | 0.190 | 1.382 (0.852–2.242) |
| Fever | **0.002** | 3.625 (1.504–8.736) | 0.153 | 1.952 (0.781–4.882) |
| Wt loss | **0.008** | 2.365 (1.233–4.536) | 0.376 | 1.399 (0.666–2.939) |
| Night sweat | 0.125 | 1.679 (0.861–3.272) | | |
| Laboratory examination | | | | |
| WBC (×10$^9$/L) | 0.133 | | | |
| Ly (×10$^9$/L) | **<0.001** | | | |
| Ne (%) | **0.002** | | | |

[a]Data are presented as numbers of patients (%) or median values (IQR) unless noted otherwise in column 1. BMI, body mass index; BCG, bacillus Calmette-Guérin; WBC, white blood cell; Ly, lymphocyte; Ne, neutrophil; REF, reference. Significant values are indicated in boldface.
[b]Silicosis categories were determined according to the 2011 revised edition of the ILO *Guidelines for the Use of the ILO International Classification of Radiographs of Pneumoconioses*.
[c]That is, including surface drillers/stone-splitters/drillers in the construction trade or quarries, plasterers, etc.

6,340/100,000 (6.34%) people, and 50.6% (491/970) of our silicosis patients were diagnosed with LTBI. Patients who have not been BCG vaccinated were found to be more vulnerable to *M. tuberculosis*. Furthermore, the QFT-GIT positivity rate was affected by age and the severity of silicosis.

**TABLE 3** Potential factors associated with latent tuberculosis infection in the study population[a]

| Variable | P | OR (95% CI) | Adj P | Adj OR (95% CI) |
|---|---|---|---|---|
| Age (median) | 0.367 | | | |
| Age <65 yrs | | REF | | REF |
| Age ≥65 yrs | 0.179 | 0.736 (0.467–1.133) | 0.370 | 0.813 (0.517–1.278) |
| | | | | |
| BMI (kg/m²) | | | | |
| <18.5 | 0.756 | 0.879 (0.477–1.583) | | |
| ≥18.5 to <24 | | REF | | |
| ≥24 to <28 | 0.489 | 1.110 (0.848–1.452) | | |
| ≥28 | 0.899 | 0.948 (0.575–1.555) | | |
| | | | | |
| Silicosis categories[b] | | | | |
| 0 | | REF | | REF |
| 1 | 0.213 | 0.710 (0.434–1.168) | 0.300 | 0.766 (0.462–1.268) |
| 2 | **0.003** | 0.470 (0.293–0.757) | **0.012** | 0.527 (0.320–0.868) |
| 3 | **0.020** | 0.589 (0.380–0.928) | 0.060 | 0.646 (0.409–1.019) |
| | | | | |
| Education | | | | |
| Illiteracy | | REF | | |
| Primary school | 0.764 | 0.946 (0.705–1.266) | | |
| Middle school or higher | 0.922 | 0.976 (0.665–1.433) | | |
| | | | | |
| Main job | | | | |
| Quarry | 0.154 | 0.606 (0.160–1.113) | | |
| Tunnel | 0.835 | 0.856 (0.379–1.871) | | |
| Others[c] | | REF | | |
| | | | | |
| Quit working | 0.823 | 0.938 (0.536–1.641) | | |
| | | | | |
| Cigarette smoking status | | | | |
| Smoker | 0.221 | 1.257 (0.898–1.764) | 0.130 | 1.314 (0.922–1.873) |
| Ex-smoker | 0.683 | 1.073 (0.777–1.483) | 0.374 | 1.160 (0.836–1.611) |
| Never smoke | | REF | | REF |
| | | | | |
| Drinking | | | | |
| Drinker | 0.204 | 1.199 (0.905–1.589) | | |
| Abstainer | 0.164 | 1.308 (0.910–1.890) | | |
| Never drink | | REF | | |
| | | | | |
| Contact of TB patients | 0.559 | 0.824 (0.429–1.581) | | |
| BCG scar | **<0.001** | 0.592 (0.456–0.769) | **0.001** | 0.791 (0.692–0.904) |

[a]Data are presented as numbers of patients (%) or median values (IQR). BMI, body mass index; BCG, bacillus Calmette-Guérin; REF, reference. Significant values are indicated in boldface.
[b]Silicosis categories were determined according to the 2011 revised edition of the ILO *Guidelines for the Use of the ILO International Classification of Radiographs of Pneumonioses*.
[c]That is, including surface drillers/stone-splitters/drillers in the construction trade or quarries, plasterers, etc.

Silicosis patients are more likely to have LTBI recurrence than the general population, making them an important source of ATB. Therefore, LTBI screening and prophylactic anti-TB treatment in patients with silicosis are of great significance for the prevention and control of TB in China. This study found that 50.6% of silicosis patients in mainland China had LTBI; their TST and QFT-GIT positivity rates were 57.2 and 40.5%, respectively. This was much higher than that of the general population in mainland China (TST positivity rate, 15 to 42%; QFT-GIT positivity rate, 13 to 20%) (9). In the LTBI screening of silicosis patients in Hong Kong, China, in 2010, the TST positivity rate was 65.9%, while that of T-SPOT.TB was 66.2%, which was even higher than that of silicosis patients in mainland China (10).

Patients with silicosis are a high-risk group for ATB, whose TB incidence is at least three times higher than that of the general population. TB has also become the leading cause of death in silicosis patients (6, 11). In this cross-sectional study, the prevalence of ATB was 6,340/100,000 (6.34%), and 21.0% of patients with silicosis had a history of

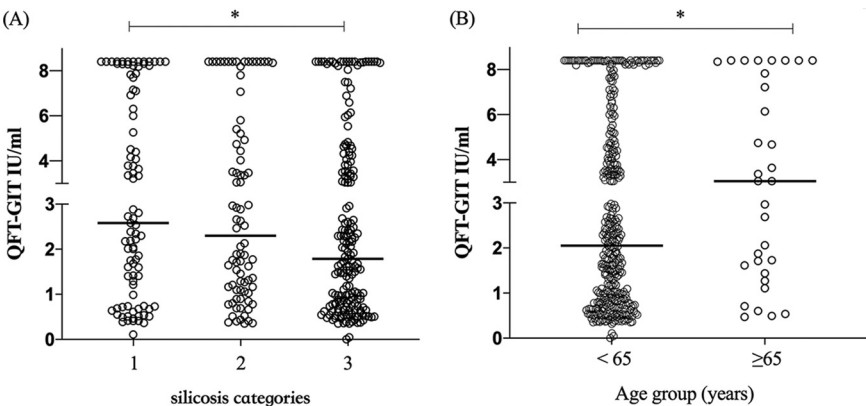

**FIG 2** Distribution of quantitative QuantiFERON-TB Gold In-Tube (QFT-GIT) results among QFT-GIT positives. (A) Distribution in different silicosis categories. (B) Distribution in different age groups.

TB. Identifying the risk factors for ATB in silicosis patients is helpful for the early identification, management, and effective prevention of TB. Previous studies have suggested that age, BMI, type of work, and silicosis categories are associated with ATB in patients (12–15). Consistent with previous studies, old age and a low BMI were associated with ATB, even though the OR was barely more than 1. Moreover, it is well known that contacts of people with tuberculosis are at high risk of developing tuberculosis (16). However, since the age group size in our study was small and the correlation was not statistically significant, it was difficult to consider age as a risk factor. Meanwhile, in the cross-sectional study, whether low BMI was a cause or a result of TB infection was not known. In contrast to previous studies, higher silicosis stage did not predict a higher risk of ATB in this study. It was worth noting that BCG vaccination proved to be a protective factor against ATB, which could reduce the risk of TB by half. In most countries and regions, the BCG vaccine must be administered within hours of birth, and its protective effect against TB has been validated (17). Since the majority of the subjects in this study were more than 50 years old and their education level was low, the history of BCG vaccination was not subjectively clear. Therefore, we used direct observation of BCG scars to determine whether BCG vaccination had been administered. Moreover, some studies have suggested that smoking is an independent risk factor for ATB, and the relationship between smoking and ATB risk in silicosis patients has been explored (18). The proportion of smokers in the ATB group was higher and the proportion of people who stopped smoking was lower than in the non-TB group, but there was no

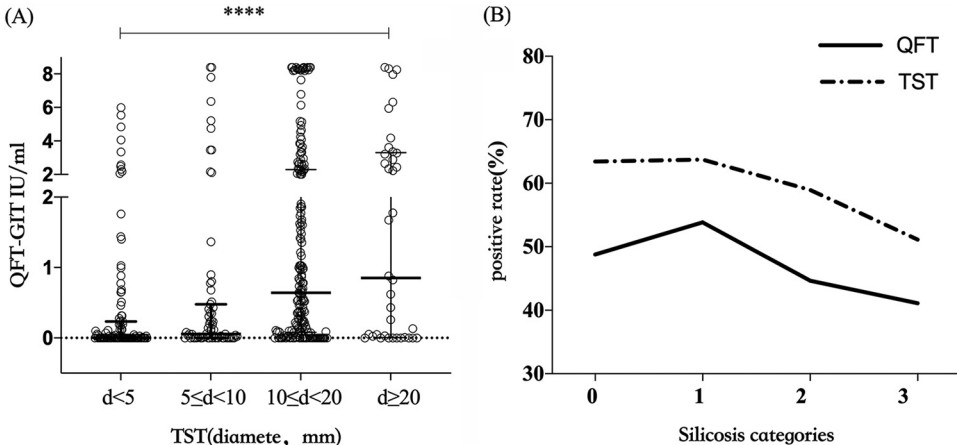

**FIG 3** Comparison of QFT-GIT and TST. (A) Differences between quantitative QFT-GIT and TST results. (B) Prevalence of LTBI determined by QFT-GIT and TST stratified by silicosis categories.

significant difference in the univariate or multivariate analyses. This may be because of the high prevalence of smoking in China. Even if the patients were not smokers, they were more likely to have been exposed to secondhand smoke in the tobacco-rich environment, which has an effect similar to that of active smoking in increasing susceptibility to ATB (19).

Interestingly, we found that the QFT-GIT positivity rate was affected by silicosis severity. The QFT-GIT positivity rate of grade 2 silicosis was lower than in grade 0 silicosis, while the risk of LTBI declined with age. Furthermore, based on the comparison of the IFN-$\gamma$ reaction intensity with QFT-GIT, the higher the severity of silicosis, the older the age, and the weaker the IFN-$\gamma$ reaction intensity. In addition, previous study showed BCG vaccination could lead to false-positive TST results (20). The consistency of TST and QFT-GIT in diagnosing LTBI was low, and the TST positivity rate was 57.2%, which was higher than that of QFT-GIT by 40.5%.

The risk factors for LTBI in patients with silicosis were explored in our study. Previous studies have shown that LTBI incidence increases with age, but neither univariate nor multivariate analyses found that the age of silicosis patients related to LTBI complications in this study, which is similar to the research results of silicosis patients in Hong Kong (10). Although an LTBI study carried out in rural Jiangsu showed that there was a limited protective effect of BCG on LTBI in general, this study found that BCG still had a significant protective effect in silicosis patients (21). It was worth exploring whether increasing stages of silicosis are associated with decreasing QFT-GIT positivity rates in univariate analysis. Therefore, this study further compared the quantitative results of QFT-GIT in patients with silicosis at various stages, which is the IFN-$\gamma$ concentration under the stimulation of TB-specific antigen, and we found that the IFN-$\gamma$ concentration decreased with the silicosis stage. Considering that the QFT-GIT detection principle is based on the immune response of lymphocytes, especially CD4$^+$ T cells, the response intensity of IFN-$\gamma$ is affected by the immune responses of lymphocytes, especially CD4$^+$ T lymphocytes (22). The immune function of silicosis patients may be abnormal owing to the deposition of silica particles and repeated pulmonary inflammatory reactions (23). To verify the immune response capacity of lymphocytes in patients with various stages of silicosis, we compared the proportions of leukocytes and lymphocytes in the peripheral blood of patients with various stages of silicosis. The results showed that the higher the stage of silicosis, the higher the number of leukocytes and the lower the proportion of lymphocytes. Meanwhile, some studies obtained lymphocytes from the bronchoalveolar lavage fluid of silicosis patients and found that apoptosis-related receptors increased by more than 20% and apoptosis-related ligands were highly expressed on CD4$^+$ T and CD8$^+$ T cells; namely, the degree of apoptosis of CD4$^+$ T and CD8$^+$ T cells increased (24). It can be seen that the higher the stage of silicosis and the lower the proportion of lymphocytes, the higher the degree of apoptosis of CD4$^+$ T cells, which may affect the immune response of CD4$^+$ T cells to TB-specific antigens, resulting in a false-negative QFT-GIT result, but further investigation of the immune mechanism is needed. Furthermore, we compared the IFN-$\gamma$ concentration of silicosis patients aged $\geq$65 years and <65 years. A lower concentration of IFN-$\gamma$ was found in the group aged $\geq$65 years, which suggests that this may be related to the decline of cellular immunity in older patients. Similarly, the IFN-$\gamma$ concentration decreased with age in a cohort of 20,486 older adults aged 50 to 70 in rural China (25). These results suggest that QFT-GIT and other immunological detection methods may have limitations in older adult patients with silicosis. In summary, the prevalence of LTBI in patients with silicosis may have been underestimated.

The present study had a few limitations. First, this is a cross-sectional study, and it is impossible to analyze the morbidity of ATB in silicosis patients statistically, so further follow-up of the incidence of ATB should be performed in this population. Second, silicosis complicated with ATB is diagnosed mainly through culture of *M. tuberculosis* in sputum and chest radiography, which may result in the misdiagnosis of some cases. Third, QFT-GIT and TST, the immunological methods used in this study, may have

false-negative results in recognizing LTBI in silicosis patients. Fourth, since some individuals may not produce scars after BCG vaccination and scars may fade over years, it was imprecise to assess BCG vaccination by looking for scars. Fifth, our study did not have individuals from Zhejiang province without silicosis as a matched control group.

In summary, 1,659 silicosis patients in Southeast China were screened for *M. tuberculosis*, and related risk factor analysis was performed. *M. tuberculosis* infection is highly prevalent in patients with silicosis. The prevalence of ATB in silicosis patients in China is as high as 6,340/100,000 (6.34%) people, and that of LTBI is 50.6%. The BCG vaccination was an independent factor that reduced the risk of *M. tuberculosis* infection in patients with silicosis. Moreover, the prevalence of LTBI in patients with silicosis may be underestimated by the QFT-GIT and TST. Screening for LTBI in this population and effective anti-TB treatment will contribute to the prevention and control of tuberculosis.

## MATERIALS AND METHODS

**Study design and participants.** This randomized controlled trial aimed to explore TB preventative treatment regimens for patients with silicosis. The trial study design has been reported in detail (http://www.clinicaltrials.gov NCT02430259 and NCT03900858) (26). A cross-sectional study was conducted among silicosis patients so that eligible participants could be invited to participate. Silicosis patients were tested for *M. tuberculosis* in the rural communities of Wenling and Sanmen Counties, Zhejiang Province, China; these areas had a high historical silicosis burden. Both counties are located in southeast China and have a TB incidence of nearly 50/100,000. Based on informed consent, all residents who had been exposed to silica dust were invited to participate in this study. The inclusion criteria were a diagnosis of silicosis, previous continuous residency at the study site for at least 1 year, and the ability to complete the investigations and tests. All participants provided written informed consent, and the study received ethical approval from the institutional review board of the First People's Hospital of Wenling, Zhejiang, China.

**Procedures.** For each eligible participant, a standard questionnaire was administered by trained interviewers via face-to-face interview. The questionnaire requested information on demographic characteristics, education, occupation, smoking history, alcohol use, history of previous TB infection, history of TB contact, and coexisting medical conditions (e.g., diabetes, hepatitis B virus infection, and immune disorders). Physicians also recorded and assessed common suspicious symptoms of pulmonary TB and silicosis and evaluated the physical examination.

Digital chest radiography was performed for all participants, and overnight collection of sputum was examined for the presence of acid-fast bacilli by smear (Ziehl-Nielsen technique) and culture. The QuantiFERON-TB Gold In-Tube test (QFT-GIT; Cellestis, Ltd., Carnegie, Victoria, Australia) and TST were used to test *M. tuberculosis* by researchers who were blinded to the characteristics of the participants. Blood samples were drawn for QFT-GIT before the TST was performed. The methodology and interpretation standards of QFT-GIT and TST have been described previously (27). A diameter of $\geq$10 mm of TST was considered positive. Venous blood samples were collected for routine blood testing.

**Definitions.** Silicosis patients were classified into four stages according to WHO guidelines. Subsequently, chest radiographs, QFT-GIT and TST results, sputum smear and culture results, and physical examination were evaluated by specialists and classified into different spectra of TB infection: (i) active TB (ATB), defined as chest radiographs showing signs of ATB and/or positive culture results; (ii) LTBI (latent TB infection) is defined by positive results for QFT-GIT or TST, with a chest radiograph that is typically normal, no sign of ATB, and negative smear and culture results; and (iii) non-TB, defined as having negative QFT-GIT and TST results, chest radiographs showing no sign of ATB disease, and negative smear and culture results.

**Statistical analyses.** Body mass index (BMI) was categorized as underweight ($<$18.5 kg/m$^2$), normal weight ($\geq$18.5 kg/m$^2$ to $<$24.0 kg/m$^2$), overweight ($\geq$24.0 kg/m$^2$ to $<$28.0 kg/m$^2$), or obese ($\geq$28.0 kg/m$^2$). A smoker was defined as an ever-smoker who was still smoking within the previous 6 months. A drinker was defined as an individual who drank alcohol regularly (more than 4 days per week). An ex-smoker/abstainer was defined as an ever-smoker/drinker who had stopped smoking/drinking for more than 6 months prior to enrollment. A never smoker was defined as an individual who had never smoked. Categorical data were compared using the *chi-squared* or *Fisher's exact* test, as appropriate. Continuous variables are summarized as the median and IQR and were compared using the *Mann-Whitney* test for unpaired data. Univariate and multivariate analyses were used to assess risk factors of *M. tuberculosis* infection, and the associations were assessed with OR and 95% CI. *Kruskal-Wallis* tests were used to compare the quantitative QFT results among different silicosis stages and age groups. Statistical analysis was performed using the statistical software GraphPad Prism (version 8.0; GraphPad Software, Inc.) and SPSS (version 23.0; IBM Corp., Chicago, IL). All *P* values were two-sided, and values of $<$0.05 were considered statistically significant.

## SUPPLEMENTAL MATERIAL

Supplemental material is available online only.
**SUPPLEMENTAL FILE 1**, PDF file, 0.1 MB.

## ACKNOWLEDGMENTS

We thank all individuals who participated in this study and the Wenling Center of Disease Prevention and Control for study site support and study enrollment.

Q.Y., M.L., and Z.H. contributed to the design of the study, directed the field work and collection of study data, as well as analysis and interpretation of the data, participated in cowriting the first draft, and approved the manuscript. X.L., Y.X., J.W., F.S., T.J., and Y.G. contributed to enrolling participants, follow-up with participants, the collection of data, analysis of the published data, and approval of the manuscript. X.H., Q.R., L.S., and W.Z. contributed to the design of the study, supervised the field trial and collection of the data, as well as analysis of the published data, and to writing and approval of the manuscript. All authors read and approved the final manuscript.

None of the authors have any financial or commercial conflicts of interest.

This study was supported by research grants from the Key Technologies Research and Development Program for Infectious Diseases of China (2017ZX10201302-004) and the Shanghai Science and Technology Committee (20DZ2210401 and 22YF1404900).

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
