## [Reviewer comments · Microbiology Spectrum]

Microbiology Spectrum

Mycobacterium tuberculosis infection among 1659 silicosis patients in Zhejiang Province, China

Qingluan Yang, Miaoyao Lin, zhangyufan he, Xuefeng Liu, Yuzhen Xu, Jing Wu, Feng Sun, Tian Jiang, Yan Gao, Xitian huang, Wenhong Zhang, Qiaoling Ruan, and Lingyun Shao

Corresponding Author(s): Qiaoling Ruan, Huashan Hospital affiliated to Fudan University

Review Timeline:

Submission Date:	April 22, 2022
Editorial Decision:	May 25, 2022
Revision Received:	July 26, 2022
Editorial Decision:	August 22, 2022
Revision Received:	October 24, 2022
Accepted:	November 9, 2022

Editor: Kileen Shier

Reviewer(s): Disclosure of reviewer identity is with reference to reviewer comments included in decision letter(s). The following individuals involved in review of your submission have agreed to reveal their identity: Robin Huebner (Reviewer #1)

Transaction Report:

DOI: <https://doi.org/10.1128/spectrum.01451-22>

May 25, 2022

Dr. Qiaoling Ruan
Huashan Hospital affiliated to Fudan University
Department of Infectious
12 M. Wulumuqi Road Shanghai 200040, China
Shanghai
China

Re: Spectrum01451-22 (Mycobacterium tuberculosis infection among 1659 silicosis patients in Zhejiang Province, China)

Dear Dr. Qiaoling Ruan:

Link Not Available

Sincerely,

Kileen Shier

Journals Department
Reviewer comments:

Reviewer #1 (Comments for the Author):

This manuscript describes the prevalence of active TB and latent TB infection in a group of Chinese patients with silicosis. This cohort is a very specific group of people, men between the ages of 53-64 who are exposed to silica. The results are hardly generalizable.

The manuscript is fairly well-written although it is confusing at times to distinguish between the different categories of TB. For example, there is ATB which is active TB, LTBI which is latent TB infection and then there is Mtb the definition of which is not clear. The authors need to clearly distinguish in the paper whether they are referring to active TB or latent infection.

The second sentence of the abstract says they want to ..."risk factors of Mtb infection and subgroups for infection control". Since

latent infection is not contagious there is no need for infection control in this group.

They provide the prevalence of active TB but give it as the number per 100,000. This is fine if you want to compare the prevalence in these patients to that of the general population but it doesn't give one any idea of the actual proportion of patients in this group with active TB. Better to also give %. And to give overall prevalence of TB in China to demonstrate how this population compares.

Were the 349 with a history of TB included in any analyses? They should have probably been included with those with active TB to assess the impact of BCG.

Direct comparisons of TST and QFT are problematic. TST picks up nontuberculous mycobacteria infection, BCG and M. tuberculosis infections. QFT is more specific and detects only Mtb.

In their analyses using TST, what size did they use? Anyone with a positive reaction or only those with > 10mm? Reactions <10 mm could be due to either BCG or NTM.

Assessing BCG vaccination by looking for scars is problematic. Some vaccinated individuals do not produce good scars and scars may fade over time, particularly in those vaccinated at birth who are now in their 50s and 60's.

Silicosis is known to be immunosuppressive. Both the TST and the QFT decrease as the silicosis category increases which may indicate immunosuppression due to the disease itself and not a lower risk of TB infection or disease or any protective effect of BCG.

Some of the odds ratios are barely over 1.0 and the p values are 0.58, 0.59. Some of the results are barely statistically significant.

The Results/Discussion section contains new data?

It is not surprising that age was not a significant variable. The age range of this group is very small, 53-64.

Multiple other studies have failed to show that BCG at birth protects against adult pulmonary TB and it has never ever been shown to protect against latent TB infection.

Reviewer #2 (Comments for the Author):

In the current manuscript, Yang and colleagues examined medical and demographic characteristics of silicosis patients from China. They found a high prevalence of tuberculosis (both active and latent TB), especially among older individuals who were contacts of TB patients. This study has numerous strengths, including its large study population, detailed methods, and robust analyses. The primary limitation is the absence of a matched control group: e.g., individuals from Zhejiang province who do not have silicosis. Controls would allow a better estimate of the community burden of TB, the relative risk of silicosis, and the protective effect of BCG vaccination.

Minor comments:

Lines 250-251: "the prevalence of LTBI in patients with silicosis may have been underestimated"

The immunological findings (e.g., lower proportion of lymphocytes at higher stages of silicosis) do support the suggestion that that QFT-GIT may generate false-negative LTBI results. However, others have suggested that BCG vaccination can lead to false-positive TST results (e.g., Lu et al. Clin Infect Dis. 2021. 72:2006-2015. doi: 10.1093/cid/ciaa519). Could BCG status explain the higher proportion of TST positivity (57.2%) compared to QFT-GIT positivity (40.5%)?

Lines 188-189: "old age and low BMI were independent risk factors for ATB". Is the association between BMI and ATB cause or effect? Is lower BMI really a risk factor for developing TB? Or is BMI lower because ATB results in weight loss? In this population, is there any association between type of work and BMI? For example, do tunnellers tend to be smaller than quarry workers?

Lines 74-76: Is the annual number of deaths from silicosis really 4 times the annual number of new cases? Ref 2 is twenty years old. Consider including a newer citation.

Table 1. Where appropriate, indicate that the numbers in parentheses represent percentages.

Reviewer #3 (Comments for the Author):

Mycobacterium tuberculosis is the name of a species which is a group of organisms with shared characteristics. It is not correct to use the term 'Mycobacterium tuberculosis' to refer to the bacterium. Manus authors choose to use the abbreviation MTB (no italics) for 'Mycobacterium tuberculosis bacteria' and then refer to MTB infection.

Lines 148-153. Please clearly define what is meant by QFT-GIT quantitative result. Is it the amount of IFN-gamma in the TB antigen tube minus the amount of IFN-gamma in the nil tube? Did the IFN-gamma response in the mitogen tube vary by age of silicosis stage?

Line 175. What TST cut-off was used to determine that 57.2% were TST-positive?

Infection with or exposure to non-tuberculosis mycobacteria (NTM) can also generate a positive TST. Were there any cases of NTM infection or disease among the persons studied?

Figure 4a. What TST cut-off was used for this Figure?

Staff Comments:

Preparing Revision Guidelines

Please return the manuscript within 60 days; if you cannot complete the modification within this time period, please contact me. If you do not wish to modify the manuscript and prefer to submit it to another journal, please notify me of your decision immediately so that the manuscript may be formally withdrawn from consideration by Microbiology Spectrum.

Spectrum01451-22R1

Title : Mycobacterium tuberculosis infection among 1659 silicosis patients in Zhejiang Province, China

Dear Prof. Kileen Shier:

Thank you for your letter and the reviewer's comments about our manuscript (Spectrum01451-22). The comments by the reviewer are constructive and helpful for improving the manuscript. We appreciate the general comments from the reviewer "*The manuscript is fairly well-written...*", "*This study has numerous strengths, including its large study population, detailed methods, and robust analyses...*". We have modified the manuscript in line with the reviewer's comments. Hereby we submit the revised manuscript for your consideration for publication. We think that we have addressed reviewer's comments to the best degree we could, and we hope this has met the reviewer's and editor's requests. Our detailed point-by-point responses to the comments are as follow:

Reviewer #1 (Comments for the Author):

1) The manuscript is fairly well-written although it is confusing at times to distinguish between the different categories of TB. For example, there is ATB which is active TB, LTBI which is latent TB infection and then there is Mtb the definition of which is not clear. The authors need to clearly distinguish in the paper whether they are referring to active TB or latent infection.

Response: We appreciate this suggestion very much. In our study, *M.tb* infection is defined as *Mycobacterium tuberculosis (M. tb)* infection status, including ATB and LTBI. For a more appropriate representation, we used MTB instead of *M.tb* infection in the revised manuscript. We had added this definition in the Introduction section (Line 96).

2)The second sentence of the abstract says they want to ..."risk factors of Mtb infection and subgroups for infection control". Since laten infection is not contagious there is no

need for infection control in this group.

Response: Thanks for your comment. We modified this sentence to “This study aimed to estimate the burden and risk factors of MTB among Chinese silicosis patients.” (Line 38).

3)They provide the prevalence of active TB but give it as the number per 100,000. This is fine if you want to compare the prevalence in these patients to that of the general population but it doesn't give one any idea of the actual proportion of patients in this group with active TB. Better to also give %. And to give overall prevalence of TB in China to demonstrate how this population compares.

Response: Thanks for your comment. We have added the percentages of active TB in lines 44,130,176,192 and 274. According to the Global Tuberculosis Report (WHO,2021), the incidence of active TB in China is 58/100,000 and Zhejiang province has a TB incidence of nearly 50/100,000, while the prevalence in silicosis patients is 6,340/100,000 in our study. We have added the data in the revised manuscript.

4)Were the 349 with a history of TB included in any analyses? They should have probably been included with those with active TB to assess the impact of BCG.

Response: We appreciate this suggestion very much. We tried to include the previous TB (349) with those with active TB (98) participants, and compared them (447) with those non-TB (479) to assess the impact of BCG. We found that having a BCG scar significantly reduced the risk of TB, with an *OR* of 0.372 (95% CI 0.281–0.498), $P < 0.001$. The result is similar to that compared ATB (98) with non-TB (479) in our manuscript, 0.320 (95% CI 0.190-0.540), $P < 0.001$.

According to our flowchart of study, we include three groups: ATB (98), LTBI (491), and non-TB patients (479) in the analysis. In order to make the study more logical and coherent, we would like to request to show only the results of the BCG comparison of ATB and non-TB.

5)Direct comparisons of TST and QFT are problematic. TST picks up nontuberculous mycobacteria infection, BCG and M. tuberculosis infections. QFT is more specific and detects only Mtb.

Response: We couldn't agree more with you. TST result is interfered by nontuberculous mycobacteria infection and BCG vaccination, but it is more economical and operated easily. QFT is more specific for *M. tb*. Even so, TST remains an important adjunct to the diagnosis of *M. tb* infection in developing countries. We believe that comparing QFT and TST can provide more information on the application of TST.

6)In their analyses using TST, what size did they use? Anyone with a positive reaction or only those with > 10mm? Reactions <10 mm could be due to either BCG or NTM.

Response: Thanks for your question and we agree with the TST reactions <10 mm could be due to either BCG or NTM. As mentioned in the manuscript, the methodology and interpretation standards of TST have been described in our previous work (Line310-311). A diameter ≥ 10 mm of TST was considered positive as we also pointed out in Line 154. We had also added the cut-off in the Materials and Methods section to clarify the definition.

7)Assessing BCG vaccination by looking for scars is problematic. Some vaccinated individuals do not produce good scars and scars may fade over time, particularly in those vaccinated at birth who are now in their 50s and 60s.

Response: We appreciate your suggestion and agree with you. We did consider this issue in the study design, however, it was very difficult to accurately to recall whether BCG vaccination was inoculated in childhood since our study subjects were 50-60 years old. After weighing the pros and cons carefully, we thought it was more accurate to use objectively existing scars to judge whether the BCG has been vaccinated. This is one of the limitations in our study. We had added it in Discussion section (Line265-267).

8)Silicosis is known to be immunosuppressive. Both the TST and the QFT decrease as the silicosis category increases which may indicate immunosuppression due to the disease

itself and not a lower risk of TB infection or disease or any protective effect of BCG.

Response: Thank you for your opinion and we agree with you very much. As we discussed in Paragraph 5 of the Discussion section, we explored the association between the stages of silicosis and QFT positivity rates as well as quantitative results. Considering that the QFT detection principle is based on the immune response, the response intensity of IFN- γ is affected by the immune responses of lymphocytes, especially CD4⁺ T lymphocytes. The immune function of silicosis patients may be abnormal owing to the deposition of silica particles and repeated pulmonary inflammatory reactions. It can be seen that the higher the stage of silicosis and the lower the proportion of lymphocytes, the higher the degree of apoptosis of CD4⁺ T cells, which may affect the immune response of CD4⁺ T cells to TB-specific antigens, resulting in a false-negative QFT-GIT result, but not a lower risk of *M. tb* infection (Line 228-249). These results suggest that QFT-GIT and other immunological detection methods may have limitations in older adult patients with silicosis.

9)Some of the odds ratios are barely over 1.0 and the p values are 0.58, 0.59. Some of the results are barely statistically significant.

Response: Thank you for your comments. When we explored the risk factors for ATB, we found that ATB was associated with age ($P=0.003$, adjusted $OR=1.059$ [95% CI 1.020–1.099]). Since this OR is barely over 1.0, we couldn't draw the conclusion that the age is statistically significant. We had modified that in Line 54, Line 61, Line195-196, and Line 272-273 (Highlighted in manuscript).

10)The Results/Discussion section contains new data?

Response: Thanks for the reminder. We had added the data in Line155-160 in Results section to explore the proportions of leukocytes and lymphocytes among different categories of silicosis.

11) It is not surprising that age was not a significant variable. The age range of this group is very small, 53-64.

Response: Thank you for your comment. Due to the small age range of the study objects, it was really difficult to conclude that whether silicosis patients of all ages have an impact on tuberculosis infection. We could only draw a conclusion that there was no statistical difference between the patients with age group of 53-64 years old.

12) Multiple other studies have failed to show that BCG at birth protects against adult pulmonary TB and it has never ever been shown to protect against latent TB infection.

Response: Thank you for your suggestion. We agree with that BCG at birth may not protect against adult pulmonary TB, as well as LTBI. However, it remains controversial. Some studies suggested that BCG vaccine had a protective effect in adult decades after vaccination and clarified the casual relationship between BCG vaccine and protection against LTBI [1,2,3]. In our study, BCG vaccination reduced the risk of ATB, with an adjusted *OR* of 0.541 (95% CI 0.307–0.954) and reduced the risk of LTBI, with an adjusted *OR* of 0.791 (95% CI 0.692–0.904). Based on our analysis and the previous studies report, we think BCG vaccination in silicosis patients may reduce the TB risk.

[1] Basu Roy, R., Sotgiu, G., Altet-Gómez, N., et al. (2012). Identifying predictors of interferon- γ release assay results in pediatric latent tuberculosis: a protective role of bacillus Calmette-Guerin? : a pTB-NET collaborative study. *American journal of respiratory and critical care medicine*, 186(4), 378–384.

[2] Chan, P. C., Yang, C. H., Chang, et al. (2013). Lower prevalence of tuberculosis infection in BCG vaccinees: a cross-sectional study in adult prison inmates. *Thorax*, 68(3), 263–268.

[3] Aronson, N. E., Santosham, M., Comstock, et al. (2004). Long-term efficacy of BCG vaccine in American Indians and Alaska Natives: A 60-year follow-up study. *JAMA*, 291(17), 2086–2091.

Reviewer #2 (Comments for the Author):

1) In the current manuscript, Yang and colleagues examined medical and demographic characteristics of silicosis patients from China. They found a high prevalence of

tuberculosis (both active and latent TB), especially among older individuals who were contacts of TB patients. This study has numerous strengths, including its large study population, detailed methods, and robust analyses. The primary limitation is the absence of a matched control group: e.g., individuals from Zhejiang province who do not have silicosis. Controls would allow a better estimate of the community burden of TB, the relative risk of silicosis, and the protective effect of BCG vaccination.

Response: Thank you for your comments. In this study, we focused on the burden and risks of tuberculosis among silicosis patients. It will be better to include individuals from Zhejiang province who do not have silicosis as matched control group, which we will mention in the limitation discussion in the revised manuscript (Line269-270).

Minor comments:

2) Lines 250-251: *"the prevalence of LTBI in patients with silicosis may have been underestimated". The immunological findings (e.g., lower proportion of lymphocytes at higher stages of silicosis) do support the suggestion that that QFT-GIT may generate false-negative LTBI results. However, others have suggested that BCG vaccination can lead to false-positive TST results (e.g., Lu et al. Clin Infect Dis. 2021. 72:2006-2015. doi: 10.1093/cid/ciaa519). Could BCG status explain the higher proportion of TST positivity (57.2%) compared to QFT-GIT positivity (40.5%)?*

Response: Thank you for your suggestions. We had added the opinion in discussion section that previous study showed BCG vaccination could lead to false-positive TST results [1] (Line 218-219). In our study, among the 409 silicosis patients for whom TST and QFT-GIT results were obtained, 188 patients had positive QFT result and 235 had positive TST result. Of the 188 QFT-positive patients, 36 had BCG scar; among the 235 TST-positive patients, 49 had BCG scar, which was not significantly different ($P=0.71$). Therefore, in our study, we thought that BCG status couldn't explain the higher proportion of TST positivity compared to QFT-GIT positivity.

[1] Lu et al. Clin Infect Dis. 2021. 72:2006-2015. doi: 10.1093/cid/ciaa519.

3) Lines 188-189: *"old age and low BMI were independent risk factors for ATB". Is the association between BMI and ATB cause or effect? Is lower BMI really a risk factor for*

developing TB? Or is BMI lower because ATB results in weight loss? In this population, is there any association between type of work and BMI? For example, do tunnellers tend to be smaller than quarry workers?

Response: Thank you for your comment. You're right. As a cross-section study, our study cannot draw the conclusion whether lower BMI was a cause or effect of ATB. As we only observed this phenomenon in our study population, we just could draw a conclusion that lower BMI was associated with ATB. We had made modification in revised manuscript. In this population, BMI wasn't associated between type of work. We compared the BMI between quarry workers and tunnellers (23.2 vs 24.1, $P=0.06$); quarry workers and other type of workers (23.2 vs 23.4, $P=0.17$); tunnellers and other type of workers (24.1 vs 23.4, $P=0.29$).

4) Lines 74-76: Is the annual number of deaths from silicosis really 4 times the annual number of new cases? Ref 2 is twenty years old. Consider including a newer citation.

Response: Thank you for your comment. According to the Global Burden of Disease study in 2019, there are more than 120,000 new cases and >7000 deaths annually.

We have updated the statistics from the Global Burden of Disease study in 2019 in line 75[4].

[4] GBD compare. IHME, University of Washington; 2019.<https://vizhub.healthdata.org/gbd-compare/>. Accessed June, 2022. This website allows the user to interrogate and download the global burden of disease data and provides a visualization tool.

5) Table 1. Where appropriate, indicate that the numbers in parentheses represent percentages.

Response: Thank you for your suggestion. We appreciate this suggestion and have added the annotation in Table 1 in the revised manuscript.

Reviewer #3 (Comments for the Author):

1) *Mycobacterium tuberculosis* is the name of a species which is a group of organisms with shared characteristics. It is not correct to use the term '*Mycobacterium tuberculosis*' to refer to the bacterium. Many authors choose to use the abbreviation MTB (no italics) for '*Mycobacterium tuberculosis* bacteria' and then refer to MTB infection.

Response: Thank you for your recommendation. We had made corrections in the manuscript, changing *M.tb* infection to MTB.

2) Lines 148-153. Please clearly define what is meant by QFT-GIT quantitative result. Is it the amount of IFN-gamma in the TB antigen tube minus the amount of IFN-gamma in the nil tube? Did the IFN-gamma response in the mitogen tube vary by age or silicosis stage?

Response: Thank you for your comments. The quantitative result of QFT-GIT is defined as the amount of IFN-gamma in the TB antigen tube minus the amount of IFN-gamma in the nil tube, which have been described in our previous work [1] (Line309-310). Since the upper limit of detection for our QFT-GIT was 8.4 IU/mL, and the mitogen tubes of almost all subjects were above 8.4 IU/mL, it was not available to compare whether there was a difference between the results of patients with different ages or silicosis stages.

[1] Yang Q, Ruan Q, Liu X, et al. Preventive tuberculosis treatment effect on QuantiFERON TB-Gold in-tube testing in a high tuberculosis-endemic country: A clinical trial. *International Journal of Infectious Diseases* 2020; 91: 182-7.

3) Line 175. What TST cut-off was used to determine that 57.2% were TST-positive?

Response: Thank you for your question. As mentioned in the manuscript, the methodology and interpretation standards of TST have been described in our previous work (Line309-310). A diameter ≥ 10 mm of TST was considered positive as we also pointed out in Line 165. We had added the cut-off in the Materials and Methods section again.

4) *Infection with or exposure to non-tuberculosis mycobacteria (NTM) can also generate a positive TST. Were there any cases of NTM infection or disease among the persons*

studied?

Response: Thank you for your comment. We strongly agree with this opinion that TST is interfered by nontuberculous mycobacteria infection. In our study, there weren't any known NTM infection cases among three groups. We examined collection of overnight sputum for the presence of acid-fast bacilli by smear (Ziehl-Nielsen technique) and mycobacteria culture, then those who had positive NTM results didn't undergo the immunological tests (QFT or TST). Therefore, NTM patients were not included in our analysis. We had added this information in Line 104-105.

5) Figure 4a. What TST cut-off was used for this Figure?

Response: Thank you for your comment. A diameter ≥ 10 mm was considered positive.

We have revised the manuscript in line with all the reviewer's comments and we hope that the manuscript is now acceptable for publication at *Microbiology Spectrum*. If you have any questions, please feel free to contact us. We appreciate your support very much.

Yours,

Qiaoling Ruan, M.D.

Department of Infectious Diseases, Fudan University

12 Wulumuqi Zhong Road, Shanghai 200040, P.R. China

Tel: 86-13661856002

Fax: +86-21-62489015

E-mail: qlruan07@fudan.edu.cn

Lingyun Shao, M.D., Ph.D.

Department of Infectious Diseases, Fudan University

12 Wulumuqi Zhong Road, Shanghai 200040, P.R. China

Tel: 86-21-52887985

Fax: +86-21-62489015

E-mail: lingyun26@fudan.edu.cn

August 22, 2022

Dr. Qiaoling Ruan
Huashan Hospital affiliated to Fudan University
Department of Infectious
12 M. Wulumuqi Road Shanghai 200040, China
Shanghai
China

Re: Spectrum01451-22R1 (Mycobacterium tuberculosis infection among 1659 silicosis patients in Zhejiang Province, China)

Dear Dr. Qiaoling Ruan:

This manuscript found four (4) correlations between risk factors in silicosis patients and active TB:

- BMI, but unclear whether this is a cause or a result of TB infection
- Patients with a known exposure to TB are more likely to have active TB (this is obvious)
- Age, but the age group size was small (53-64) and the correlation was not statistically significant
- BCG vaccination, which is the only meaningful correlation based upon the data presented in this manuscript

The study is interesting because of the unique population and study size of silicosis patients. However, the conclusions drawn for the first three points above lack supporting data. I would be willing to consider a revised manuscript with emphasis on the BCG vaccination. Alternatively, the authors may consider submitting to another journal that will accept the manuscript as is.

Minor comments:

Line 57: Change to MTB here and search throughout manuscript for other occurrences of "M. tb infection"

Line 61-62: As pointed out by a reviewer, is low BMI a cause or an effect of ATB? If you don't know that it is a cause then you cannot say that patients with low BMI are vulnerable to ATB.

Line 75: Does China no longer have the highest incidence of silicosis? That's what this sentence implies.

Line 76: Has the situation improved? It seems like deaths have gone down (>24,000 to >7,000) but new cases have skyrocketed (6,000 new cases to >120,000 new cases).

Line 105: "those who had non-tuberculosis mycobacteria infection."

Line 192: 21.0% had a history of TB-how was this identified?

Line 200: It makes sense that exposure to patients with TB is a risk factor for ATB, but I don't think the explanation of a weakened immune system makes sense in this context.

Link Not Available

Sincerely,

Kileen Shier

Journals Department
Reviewer comments:

Staff Comments:

Preparing Revision Guidelines

Please return the manuscript within 60 days; if you cannot complete the modification within this time period, please contact me. If you do not wish to modify the manuscript and prefer to submit it to another journal, please notify me of your decision immediately so that the manuscript may be formally withdrawn from consideration by Microbiology Spectrum.

Spectrum01451-22R2

Title : Mycobacterium tuberculosis infection among 1659 silicosis patients in Zhejiang Province, China

Dear Prof. Kileen Shier:

Thank you for your letter about our manuscript (Spectrum01451-22R2). The comments are constructive and helpful for improving the manuscript. We appreciate the general comments "*The study is interesting because of the unique population and study size of silicosis patients*". We have modified the manuscript in line with your comments. Hereby we submit the revised manuscript for your consideration for publication. We think that we have addressed your comments to the best degree we could, and we hope this has met your requests. Our detailed point-by-point responses to the comments are as follow:

This manuscript found four (4) correlations between risk factors in silicosis patients and active TB:

•BMI, but unclear whether this is a cause or a result of TB infection

•Patients with a known exposure to TB are more likely to have active TB (this is obvious)

•Age, but the age group size was small (53-64) and the correlation was not statistically significant

•BCG vaccination, which is the only meaningful correlation based upon the data presented in this manuscript

The study is interesting because of the unique population and study size of silicosis patients. However, the conclusions drawn for the first three points above lack supporting data. I would be willing to consider a revised manuscript with emphasis on the BCG vaccination. Alternatively, the authors may consider submitting to another journal that will accept the manuscript as is.

Response: Thanks for your suggestion, we really appreciate your positive feedback and we agree with you that we didn't know the cause-and-effect relationship between BMI and TB infection thus we have removed BMI as a risk factor in the discussion part. Also, it's apparent that close contact with ATB patients will increase the risk of developing TB, therefore, we didn't underline it anymore. In addition, considering age as a risk factor also lacking strong evidence since the age group size was small, we have emphasized on the BCG vaccination in our revised manuscript and draw a conclusion that the BCG vaccination was an independent factor that reduced the risk of MTB in patients with silicosis. As marked in 52-54, 60-61, 174-176, 276-279 we removed the age and low BMI from the conclusion and in Line 196-203, we revised as "It is well-known that contacts of people with tuberculosis are at high risk of also developing tuberculosis. Since the age group size in our study was small and the correlation was not statistically significant, it was difficult to consider age as a risk factor. Meanwhile, in the cross-sectional study, whether low BMI was a cause or a result of TB infection was not known.....It was worth noting that BCG vaccination proved to be a protective factor against ATB". In addition, since we no longer emphasize the factors of age and weight, we have deleted Figure 2 to make the article clearer. This comment is helpful for improving the manuscript and thank you very much.

Minor comments:

Line 57: Change to MTB here and search throughout manuscript for other occurrences of "M. tb infection"

Response: We really appreciate your kind reminding, we have searched throughout our manuscript and changed all "M.tb infection" to MTB in line 52, line 56, line 59, line 62, line 104 and line 265 in our revised manuscript.

Line 61-62: As pointed out by a reviewer, is low BMI a cause or an effect of ATB? If you don't know that it is a cause then you cannot say that patients with low BMI are vulnerable to ATB.

Response: We agree with you and have removed low BMI as a risk factor in the infection of tuberculosis in line 52, line 60, line 174 and line 277 by reworded the description as the following: in the cross-sectional study, whether low BMI was a cause or a result of TB infection was not known.

Line 75: *Does China no longer have the highest incidence of silicosis? That's what this sentence implies.*

Response: No, actually, China still has the highest incidence of silicosis. Twenty years ago, China has the highest incidence of silicosis, with 6000 new cases and >24 000 deaths reported annually. There are still more than 120,000 new cases and >7000 deaths annually, and we have revised the sentence in line 75 which was described as “There are still more than 120,000 new cases and >7000 deaths annually”.

Line 76: *Has the situation improved? It seems like deaths have gone down (>24,000 to >7,000) but new cases have skyrocketed (6,000 new cases to >120,000 new cases).*

Response: We appreciate your questions and comments. The situation didn't improve compared to twenty years ago, China still has the highest incidence and deaths, with >12,000 deaths and >130,000 new cases globally.

Line 105: *"those who had non-tuberculosis mycobacteria infection."*

Response: We apologize for the confusing and we have corrected it in our revised manuscript in line 104 as suggested.

Line 192: *21.0% had a history of TB-how was this identified?*

Response: Thanks for your question. For each eligible participant, a standard questionnaire was administered by trained interviewers via face-to-face interview. Participants who had the diagnosis proof of tuberculosis previously or had undergone effective anti-tuberculosis treatment were identified as having a history of tuberculosis.

Line 200: It makes sense that exposure to patients with TB is a risk factor for ATB, but I don't think the explanation of a weakened immune system makes sense in this context.

Response: Thank you for your recommendation and we agree with you that a weakened immune system couldn't explain the development of ATB, thus we have rephrased the sentence in our revised manuscript in line 201 as the following: In addition, contacts of people with tuberculosis are at high risk of also developing tuberculosis.

We have revised the manuscript in line with all the reviewer's comments and we hope that the manuscript is now acceptable for publication at *Microbiology Spectrum*. If you have any questions, please feel free to contact us. We appreciate your support very much.

Yours,

Qiaoling Ruan, M.D.

Department of Infectious Diseases, Fudan University

12 Wulumuqi Zhong Road, Shanghai 200040, P.R. China

Tel: 86-13661856002

Fax: +86-21-62489015

E-mail: qlruan07@fudan.edu.cn

Lingyun Shao, M.D., Ph.D.

Department of Infectious Diseases, Fudan University

12 Wulumuqi Zhong Road, Shanghai 200040, P.R. China

Tel: 86-21-52887985

Fax: +86-21-62489015

E-mail: lingyun26@fudan.edu.cn

November 9, 2022

Dr. Qiaoling Ruan
Huashan Hospital affiliated to Fudan University
Department of Infectious
12 M. Wulumuqi Road Shanghai 200040, China
Shanghai
China

Re: Spectrum01451-22R2 (Mycobacterium tuberculosis infection among 1659 silicosis patients in Zhejiang Province, China)

Dear Dr. Qiaoling Ruan:

Your manuscript has been accepted, and I am forwarding it to the ASM Journals Department for publication. You will be notified when your proofs are ready to be viewed.

Sincerely,

Kileen Shier
Editor, Microbiology Spectrum
